# The Biological Effects of Complete Gasoline Engine Emissions Exposure in a 3D Human Airway Model (MucilAir^TM^) and in Human Bronchial Epithelial Cells (BEAS-2B)

**DOI:** 10.3390/ijms20225710

**Published:** 2019-11-14

**Authors:** Pavel Rossner, Tereza Cervena, Michal Vojtisek-Lom, Kristyna Vrbova, Antonin Ambroz, Zuzana Novakova, Fatima Elzeinova, Hasmik Margaryan, Vit Beranek, Martin Pechout, David Macoun, Jiri Klema, Andrea Rossnerova, Miroslav Ciganek, Jan Topinka

**Affiliations:** 1Department of Genetic Toxicology and Nanotoxicology, Institute of Experimental Medicine of the CAS, Videnska 1083, 142 20 Prague, Czech Republic; tereza.cervena@iem.cas.cz (T.C.); kristyna.vrbova@iem.cas.cz (K.V.); antonin.ambroz@iem.cas.cz (A.A.); zuzana.novakova@iem.cas.cz (Z.N.); fatima.elzeinova@iem.cas.cz (F.E.); hasmik.margaryan@iem.cas.cz (H.M.); andrea.rossnerova@iem.cas.cz (A.R.); jan.topinka@iem.cas.cz (J.T.); 2Department of Physiology, Faculty of Science, Charles University, Vinicna 7, 128 44 Prague, Czech Republic; 3Center of Vehicles for Sustainable Mobility, Faculty of Mechanical Engineering, Czech Technical University in Prague, Technicka 4, 160 00 Prague, Czech Republic; michal.vojtisek@fs.cvut.cz (M.V.-L.); vit.beranek@fs.cvut.cz (V.B.); 4Department of Vehicles and Ground Transport, Czech University of Life Sciences in Prague, Kamycka 129, 165 21 Prague, Czech Republic; pechout@tf.czu.cz (M.P.); macound@tf.czu.cz (D.M.); 5Department of Computer Science, Czech Technical University in Prague, 12135 Prague, Czech Republic; klema@fel.cvut.cz; 6Department of Chemistry and Toxicology, Veterinary Research Institute, 621 00 Brno, Czech Republic; ciganek@vri.cz

**Keywords:** complete engine emissions, gene expression, 3D models, cell monocultures

## Abstract

The biological effects induced by complete engine emissions in a 3D model of the human airway (MucilAir^TM^) and in human bronchial epithelial cells (BEAS-2B) grown at the air–liquid interface were compared. The cells were exposed for one or five days to emissions generated by a Euro 5 direct injection spark ignition engine. The general condition of the cells was assessed by the measurement of transepithelial electrical resistance and mucin production. The cytotoxic effects were evaluated by adenylate kinase (AK) and lactate dehydrogenase (LDH) activity. Phosphorylation of histone H2AX was used to detect double-stranded DNA breaks. The expression of the selected 370 relevant genes was analyzed using next-generation sequencing. The exposure had minimal effects on integrity and AK leakage in both cell models. LDH activity and mucin production in BEAS-2B cells significantly increased after longer exposures; DNA breaks were also detected. The exposure affected *CYP1A1* and *HSPA5* expression in MucilAir^TM^. There were no effects of this kind observed in BEAS-2B cells; in this system gene expression was rather affected by the time of treatment. The type of cell model was the most important factor modulating gene expression. In summary, the biological effects of complete emissions exposure were weak. In the specific conditions used in this study, the effects observed in BEAS-2B cells were induced by the exposure protocol rather than by emissions and thus this cell line seems to be less suitable for analyses of longer treatment than the 3D model.

## 1. Introduction

Road traffic is one of the major sources of air pollution, particularly in large cities. Human exposure to traffic-related pollutants is associated with increased cardiopulmonary mortality, adverse reproductive effects, incidence of neurodegenerative disorders, and cancer [1]. Traffic pollutants consist of a complex mixture of gaseous and solid components, including carbon monoxide, carbon dioxide, nitrogen oxides, volatile organic compounds, polycyclic aromatic hydrocarbons, heavy metals, secondary reaction products, and particulate matter [2]. The International Agency for Research on Cancer (IARC) has classified diesel engine emissions as carcinogenic to humans (Group 1), and gasoline engine emission as a possible carcinogen to humans (Group 2B) [3]. Due to the negative health impacts of traffic emissions, the evaluation of biological consequences of exposure to these mixtures is of great importance. Toxicological tests have for many years relied on the treatment of cellular monocultures of various origins and properties, including both bacteria and eukaryotic cells [4,5]. Such monocultures have become part of a battery of tests recommended by the OECD. However, these systems have numerous disadvantages, including limited genetic stability if cancer or immortalized cell lines are used, or a lack of inter- and intracellular signaling. This affects various cellular functions, including gene expression regulation, cell cycle or apoptotic response. As a result, cellular monocultures may not represent an ideal system to evaluate the toxicity of environmental pollutants. While tests in experimental animals are also commonly performed, ethical issues and anatomical and biochemical differences between the animals and humans make them less suitable systems for analyses of toxicity. In recent years, 3D cultures have become an emerging platform of toxicity assessment [6]. Various forms of these systems exist, including co-cultures consisting of two or more different cell types that mimic inter- and intracellular interactions found in in vivo conditions. However, they lack the typical 3D structure of organs/tissues. Therefore, 3D reconstructed tissue models originating from real human tissues of various origin are a better alternative. The models mimicking skin are most commonly used, but lung tissues for the purpose of inhalation toxicology are also available [7,8].

Depending on the endpoints to be assessed, in vitro toxicity tests of vehicle emissions may be carried out in various settings. They include for example the exposure of the test system to complete emissions [9,10,11,12,13,14], suspended particles [15,16,17,18], or to extracts from particles [19,20,21,22]. Each of these approaches has its specific advantages and limitations. In general, the exposure to extracts is less technically demanding as no specific exposure systems are required, but the output of such experiments is limited as it only reflects the effect of compounds dissolved in the solvent used to prepare the extracts. Therefore, extractable organic matter (EOM), obtained from particulate matter extracted using dichloromethane, contains various organic compounds including polycyclic aromatic hydrocarbons (PAHs), the compounds responsible for most of the mutagenic activities of EOM. However, such extracts do not reflect the effect of particles or gaseous pollutants, including nitric oxide or nitrogen dioxide. Although experimental approaches that use suspended particles address some of the disadvantages associated with the exposure to extracts, they still have certain limitations including difficulties with the isolation of particles in their original form (particularly their size) and the lack of the gaseous components of the emissions. To account for the effect of these pollutants, exposures to complete emissions are needed. Thus, exposure systems that allow the application of complete emissions represent the most realistic settings that mimic real-world exposure scenarios.

The time distribution of emissions exposure is another aspect that should be considered when the biological effects of traffic pollutants are evaluated. As real-life exposures are usually long-term, with periods of higher levels and lower concentrations of air pollutants, such scenarios should ideally be mimicked in toxicological experiments. While animal studies are commonly designed as long-term, lasting for several weeks or even months [23], experiments in vitro with cellular monocultures are usually short-term, limited to the exposure of hours up to single days (e.g., [17,21]), although in one study a 6-month treatment was performed [24]. However, exposure to complete emissions was not seen in any of these studies. Unlike cellular monocultures, 3D models have a greater potential for long-term exposure studies [25]. Regardless of this, the experiments with complete emissions reported so far have all been limited to short-term treatments, usually not exceeding 6 h [10,12,14,26,27].

In this study we aimed to compare the biological response in two different model systems: BEAS-2B cells, a typical cellular monoculture used in toxicological tests, and MucilAir^TM^, a commercial 3D model originating from bronchial epithelial tissue consisting of human basal, goblet, and ciliated cells [8]. Both systems were grown at the air–liquid interface (ALI) and were intermittently exposed to complete emissions for up to five days in a recently developed exposure system [28]. The different response of the systems was evaluated using transepithelial electrical resistance (TEER), mucin production and cytotoxicity measurements, detection of DNA breaks, and analysis of expression of the selected 370 relevant genes.

## 2. Results

### 2.1. TEER Measurement, Mucin Production, and Cytotoxicity Analyses

Transepithelial electrical resistance (TEER) was measured before and after the exposure to complete emissions as described in detail in Section 4.3. The TEER values for all BEAS-2B cell samples were around 200 Ω*cm^2^ and no changes were observed during exposure to the emissions. For the MucilAir^TM^ tissues, a significant drop of TEER was found after one-day exposure for both the exposed and control samples when the T1 and T0 time points were compared (Figure 1A). However, all TEER values were still well above the minimal acceptable value of 200 Ω*cm^2^, suggesting that the exposure had no negative impacts on tight junctions. Interestingly, no significant effects of the complete emissions exposure on TEER were observed at time point T1 (a comparison of the exposed and control samples at this time point). For five-day exposure, significant differences between TEER values measured at T2, T3, T4, and T5 vs. T0 were found (Figure 1B). Apart from time point T5, these differences were limited to the exposed samples. At time points T2 and T4, significantly higher TEER was observed for the exposed samples than for the controls. Overall, the TEER measurement results indicate that exposure to the complete emissions and/or the clean air had a minimal effect on the integrity of the MucilAir^TM^ model.

As reported in Section 4.3, mucin production was assessed in a culture media obtained before and after exposure to complete emissions. Mucin production by the MucilAir^TM^ tissues was high at time point T0 and tended to decrease at time points T1 and T2. After the one-day treatment, significantly lower mucin levels were found in the control samples when compared with time point T0; the decrease in the exposed did not reach statistical significance (Figure 2A). For five-day exposure, a non-significant trend of mucin production increase after time point T3 was found. This increase was more pronounced for the control samples, and at time point T4 a significant difference between the exposed and control tissues was detected (Figure 2B). In contrast, in BEAS-2B cells the production was very low after one-day exposure and no differences between the exposed and control samples were observed (Figure 2C). The five-day treatment was characterized by a time-dependent increase in mucin levels that was significant for both the exposed and control samples at time points T3, T4, and T5, when compared with T0. At time point T5, mucin production was significantly elevated in the controls when compared with the exposed samples (Figure 2D).

Measurements of lactate dehydrogenase (LDH) and adenylate kinase (AK) activity in culture medium were used as markers of cytotoxicity. Similarly to the TEER and mucin, both the LDH and AK were analyzed in samples collected before and after exposure to complete emissions. The activity of AK was generally very low (less than 1%) in both cell models (Figure 3A–D). It only exceeded 2% in BEAS-2B cells after five-day exposure (time points T4 and T5, Figure 3D) suggesting a weak effect of the complete emissions on AK leakage after the longer treatment. The activity of LDH followed a similar trend, but the response was more pronounced (Figure 4A–D). While no effects of exposure were found in the MucilAir^TM^ system (Figure 4A,B), in BEAS-2B cells the one-day treatment increased the LDH leakage up to 10.9% in the exposed cells and 16.1% in the controls (Figure 4C). The five-day exposure was characterized by an almost linearly increasing LDH activity that exceeded 55% in both the exposed and the control samples at time point T5 (Figure 4D). From time point T2, this increase was statistically significant when compared with the onset of treatment (T0). No consistent differences between the exposed and controls for individual time points were found, suggesting a minor role of exposure to the complete emissions in LDH leakage.

### 2.2. Detection of DNA Breaks

Double-strand DNA breaks, analyzed as phosphorylation of histone H2AX, were assessed in cell lysates prepared from cells collected at time points T1 (after one-day exposure) and T5 (after five-day exposure). In the MucilAir^TM^ tissue, exposure to the complete emissions did not induce phosphorylation of histone H2AX: after the five-day treatment, the levels were comparable for the exposed and controls, while after the one-day exposure, higher DNA damage was found in the controls (Figure 5A). In BEAS-2B cells, exposure to complete emissions significantly increased the levels of double-strand DNA breaks after both time intervals (Figure 5B). Interestingly, phosphorylation of histone H2AX was lower after the longer exposure periods than after the one-day treatment in both cell models.

### 2.3. mRNA Expression Analysis

The differences in mRNA expression between the exposed and control samples, induced by complete emissions, were generally very small (Table 1). In the MucilAir^TM^ tissues, the expression of CYP1A1 (cytochrome P450 family 1 subfamily A member 1) was induced in both exposure periods. The one-day treatment also caused a reduction of HSPA5 (heat shock protein family A (Hsp70) member 5) expression. In BEAS-2B cells, no significant differences were found after either one-day or five-day treatments.

To investigate the possible adverse effects of prolonged, five-day incubations, gene expression at time points T1 and T5 was compared. For the MucilAir^TM^ tissue, the differences were found for the exposed samples (decreased expression of SLC2A3 (solute carrier family 2 member 3) and AQP4 (aquaporin 4)), but not for the controls (Table 2) suggesting that this model is not impacted by the processes associated with handling the inserts and growing the cells at the ALI for an extended period of time. In contrast, in BEAS-2B cells, such a comparison yielded a substantially greater number of deregulated genes: for the exposed samples, 56 genes were differentially expressed, while for the controls the difference was observed for 52 transcripts (Appendix A). Among these genes, 33 were commonly deregulated, 22 were expressed in the exposed samples, and 18 were expressed in controls only (Appendix A). The genes involved in apoptosis/necrosis constituted almost one third of the common transcripts, suggesting negative effects of prolonged, five-day exposure on the general state of BEAS-2B cell cultures.

We then focused on the comparison of gene expression between both model systems. As expected, substantial differences were observed: 165, 175, 169, and 172 significantly differentially expressed genes were found for a comparison between MucilAir^TM^ and BEAS-2B cells at T1 and T5 time points for the exposed and controls, respectively (Appendix A). Further analysis revealed that most of these differentially deregulated genes were common, regardless of the time of exposure (one or five days), sample type (exposed or controls), and model system (MucilAir^TM^ or BEAS-2B) (Appendix A). These genes were involved in all of the biological pathways investigated in this study (described in Section 4.8), which highlights the importance of the selection of the cellular model for toxicological tests and suggests that caution needs to be taken when interpreting the data, as they strongly depend on the properties of the model system.

## 3. Discussion

This study was designed to evaluate the actual effects of exposure to complete emissions in two cell systems growing at the ALI: a 3D model of human lung tissue and a standard cell monoculture commonly used in toxicological tests. The exposure included a combination of treatment with complete emissions and filtered air in the exposure system preceded by and (in the case of five-day exposure) followed by incubation in a standard cell incubator. To monitor the impact of exposure to the emissions, as well as the effect of the environmental change (the cell cultures were transferred repeatedly between the cell incubator and the exposure system), the characteristics describing the overall state of the samples and cytotoxicity of the extracts were continuously assessed at selected time points. The genotoxic effects of the exposure were evaluated by measuring double-strand DNA breaks and by differential analysis of gene expression.

The TEER measurement, which is used to evaluate tissue integrity and tight junctions, was checked before the cells were transported from the tissue culture laboratory to the exposure facility and then after each exposure unit had finished. In MucilAir^TM^, some effects of exposure were clearly visible, as TEER values significantly differed between time point T0 and measurements taken during the treatment. However, it is important to note that TEER values were always well above 200 Ω*cm^2^, suggesting that there were no negative biological effects of complete emissions exposure and samples handling on tissue integrity of MucilAir^TM^. For BEAS-2B cells, no changes of TEER were detected regardless of the time of exposure; the values remained around 200 Ω*cm^2^. This result is in line with the study by Stewart et al. in which TEER values in various bronchial epithelial cell systems were compared [29]. The authors concluded that BEAS-2B cells do not differentiate at the ALI and do not develop tight junctions. This fact limits the suitability of this cell line as a model system for complete emissions exposure.

Mucins, constituents of mucus, play a very important role in the protection of airways against negative environmental factors and their production increases in response to inflammatory stimuli [30]. It has been demonstrated that repeated exposure of 3D human bronchial tissue cultures to whole cigarette smoke affects mucins production [31]. In another study, intestinal mucins protected cell cultures against DNA double-strand breaks induced by a genotoxic agent [32]. Our results for BEAS-2B cells indicate that exposure to complete emissions induces mucin production. In these cells, basal mucin levels are very low but they already gradually increase after two days of exposure. Interestingly, as the differences between the exposed and control samples were minor, the response is probably not related to the presence of pollutants in the engine exhaust, but is more likely a protective reaction of the cells against unfavorable conditions associated with changes of the environment during the exposure itself. Mucin production by MucilAir^TM^ is in general many times higher than that in BEAS-2B cells. However, unlike the cells, mucin production tends to decrease after exposure starts and then it slowly increases to the basal levels. Increased mucin production was not observed for MucilAir^TM^ suggesting that this system copes better with incubation conditions and underlines its suitability for long-term exposure tests.

For the detection of cell membrane damage, a parameter of cytotoxicity, adenylate kinase and lactate dehydrogenase leakage, was assessed. The activity of AK was very low in all MucilAir^TM^ samples. For BEAS-2B cells, a non-significant increase of AK levels was detected, starting on day three (time point T3) of five-day exposure. The activity of LDH followed a similar trend, although the changes detected in BEAS-2B cells were more pronounced (cytotoxicity reaching up to 75%) and already significant after the second day of five-day exposure. Again, no consistent differences between the exposed and control samples were found. These observations support the TEER measurement and mucin production data and confirm that BEAS-2B cells are not an optimal system for exposure monitoring at the ALI.

The detection of histone H2AX phosphorylation is a commonly used marker of double-strand DNA damage, although γ-H2AX formation is also an early indicator of apoptosis [33]. In previous studies, diesel exhaust particles (DEP) induced double-strand DNA breaks in A549 and BEAS-2B cells [34,35], while no effects of chronic (6 month) exposure of BEAS-2B cells to DEP were found in another report [24]. However, to date, γ-H2AX formation has not been investigated following exposure to complete emissions. Our results indicate no effects of such exposure on MucilAir^TM^ for both one-day and five-day treatment. In BEAS-2B cells, a significant induction of double-strand DNA damage was found for both exposure intervals suggesting the genotoxic effects of complete emissions in this cell model, and again showing greater sensitivity of this monoculture when compared with the 3D model.

For a more detailed evaluation of the biological effects induced by complete emissions exposure, we performed gene expression analysis. Specifically, 370 selected relevant genes involved in the processes presumably affected by engine exhaust exposure were analyzed. Gene expression modulation by complete engine exhaust (diesel or gasoline) was previously investigated by several authors, although these studies focused on a limited number of genes. Thus, the effect of complete diesel exhaust from engines with and without particulate filter using traditional fuel and biodiesel was assessed in human bronchial epithelial cells NHBE grown at the ALI. The cells were exposed to diluted exhaust (diluted 1:20) for 5, 20, or 60 min, and 2 h after the exposure the gene expression was analyzed. The expression of *HO-1* and *CYP1A1* was induced after 60-min exposure; the filtered exhaust caused more pronounced effects [11]. In the triple-cell co-culture system, the effect of 2-h and 6-h exposure to complete diesel exhaust emissions, followed by a 6-h post-incubation period, on the expression of 84 genes associated with DNA repair, apoptosis, and cell cycle regulation, was investigated. Changes of gene expression were observed for a number of genes in all three groups [36]. The exposure to complete gasoline exhaust from engines with and without particle filters and using various fuels for 6 h followed by a 6-h post-incubation period had no effect on oxidative stress and inflammation-related genes in a multicellular human lung model and in MucilAir^TM^. However, the repeated 3 × 6 h exposure to an exhaust from an engine without the filter, followed by post-incubation periods, increased the expression of *HMOX1* and *TNFα* genes in the multicellular model suggesting that longer exposure is required to induce the biological effects [14]. The effect of ethanol-containing fuels was tested in the multi-cellular human lung model consisting of three cells types. The cells were exposed for 6 h, followed by a 6-h post-incubation to the exhaust from fuels containing 10%, 85% ethanol, pure gasoline, and diesel fuel. Although there were no changes in the expression of oxidative stress- and inflammation-related genes observed after exposure to gasoline exhaust, diesel exhaust induced the expression of *HMOX1*, *GSR*, and *IL-8* [27]. Finally, the exposure to complete gasoline exhaust for 2 × 6 h during a 48-h incubation had no effect on the expression of oxidative stress- and inflammation-related genes in a co-culture of lung cells, monocytes, and dendritic cells growing at the ALI [37]. Overall, these studies indicate that the gene expression is induced by exposure to diesel exhaust, while gasoline emissions have hardly any effect.

In this study, the gene expression changes induced by an ethanol-containing gasoline engine exhaust were weak in both cellular models. In BEAS-2B cells, no induction of gene expression was observed either after one day or five days of exposure. The reason for the lack of effect is not clear. We may speculate that unfavorable incubation conditions, manifested by an increased cytotoxicity and impacting mucin production, had a greater impact on the gene expression that suppressed the manifestation of the possible effects of engine emissions. In MucilAir^TM^, the elevated expression of *CYP1A1* was detected in both time periods. The protein encoded by this gene is involved in the metabolism of many compounds, including polycyclic aromatic hydrocarbons. In our previous study, the expression of *CYP1A1* in MucilAir^TM^ was affected by exposure to benzo[a]pyrene, although the effects were detected after longer (7 and 28 days) exposure periods [25]. Therefore, the deregulation of *CYP1A1* expression probably reflects the presence of PAHs in the exhaust. The one-day exposure further decreased *HSPA5* expression. This gene encodes heat shock protein family A (Hsp70) member 5, a protein that is involved in the folding and assembly of endoplasmic reticulum proteins. This observation suggests the negative impacts of exposure on the folding and transport of proteins in the cell, as previously observed in primary bronchial epithelial cells exposed for 6 h to complete diesel exhaust followed by 3 h post-incubation [38].

To reveal the potential effects of a prolonged, five-day incubation at the ALI, the gene expression of samples incubated for 5 days was compared with those exposed for 1 day. In the control MucilAir^TM^ samples, the gene expression at time points T5 and T1 was comparable; in the exposed tissues, a decreased gene expression was found for *SLC2A3* (solute carrier family 2 member 3; the protein plays a role in monosaccharide transport across the membrane) and *AQP4* (aquaporin 4; the protein is important in keeping brain water homeostasis). This suggests that the incubation and handling of the samples had minimal effects on the MucilAir^TM^ tissues. In contrast, in BEAS-2B cells the exposure was associated with modulation of the expression of a number of genes, among which 33 were commonly deregulated in exposed and control samples. These genes represent a cellular response to five-day incubation, regardless of the exposure to engine emissions or to clean air. Among other processes the genes included those involved in apoptosis, suggesting that the longer incubation of BEAS-2B cells had negative impacts on this cell model. This is in agreement with the above-mentioned observations for mucin and cytotoxicity assessment. Moreover, histone H2AX phosphorylation, detected in this cell line, may indicate the induction of apoptotic response, rather than DNA damage induced by complete emissions.

The type of the cellular model was the most prominent factor affecting gene expression. We identified 117 common genes that differed when model systems (MucilAir^TM^, BEAS-2B), type of sample (exposed, controls) and time of exposure (one day, five days) were compared. While these genes were involved in all of the 13 biological pathways investigated in this study, almost 30% (32 of 117) of them participated either in mitochondrial energy metabolism, DNA damage and repair, or cytochrome P450s and phase I drug metabolism. These processes may be regarded as key mechanisms that differentiate both cell systems in terms of their response to the type and duration of treatment. The differences may fundamentally affect the levels of studied parameters, as well as the interpretation of the results. Therefore, the selection of the model system should be carefully taken into account when planning the toxicological tests.

In summary, our study showed that the exposure to the complete emissions generated by an engine running on a blend of ethanol and ordinary gasoline, induced minimal biological effects after one-day and five-day treatments in both MucilAir^TM^ tissues and BEAS-2B cells. Further analyses confirmed the suitability of the 3D system for such tests and highlighted the limited use of the standard cell monoculture in complete emissions exposure settings. While generally weak, the biological response of the 3D model suggests low health risks associated with exposure to emissions produced by modern gasoline engines, yet more tests need to be performed to confirm this statement. Specifically, detailed whole genome gene expression analyses should be conducted and exposure periods for MucilAir^TM^ tissues should be extended for up to several weeks. Although in this study the expression of toxicologically-relevant genes was analyzed, we cannot rule out that other processes not covered by the targeted panel were affected.

## 4. Materials and Methods 

### 4.1. Cell Cultures

The effects of complete emissions were tested in two cell models: BEAS-2B, human bronchial epithelial cells (CRL-9609TM, ATCC^®^, Manassas, VA, USA) and MucilAir^TM^, 3D lung tissue model (Epithelix Sàrl, Geneva, Switzerland). The BEAS-2B cells are an adherent cell line derived from lung autopsy of a healthy man, immortalized by a virus [39] that exhibit standard morphology and metabolism if grown under recommended conditions. MucilAir^TM^ is a fully differentiated bronchial epithelial 3D model reconstituted from primary human cells from healthy donors. Cell inserts consist of human basal, goblet, and ciliated cells cultured at an air–liquid interface, representing a fully differentiated and functional respiratory epithelium. Unlike the standard cell lines, MucilAir^TM^ does not overgrow after longer cultivation but it still retains the ability to heal when damaged [8]. The model displays in vivo characteristics including stratification, tight junctions, metabolic activity, mucus production and cilia beating. The samples evaluated in our study were obtained from a 64-year old Caucasian female, non-smoker, with no pathology reported. The cell models were grown at ALI at 37 °C, 5% CO_2_, and relative humidity >90% in 24-well format Transwell^®^ cell culture inserts (Sigma-Aldrich, St Louis, MO, USA). For BEAS-2B cells, serum-free cultivation conditions (BEGM™ kit CC- 3170; Lonza, Basel, Switzerland) were used and 100,000 cells/insert were seeded [40]. Twenty-four hours after seeding, the apical medium was removed from the insert and the cells were kept 24 h at ALI conditions prior to exposure. Using this procedure, we were able to prepare inserts with BEAS-2B monolayer in which no medium leakage from the basal area occurred. For 3D models, MucilAir^TM^ culture medium (Epithelix Sàrl, Geneva, Switzerland) was used and the cultures were monitored for 3 weeks to determine their stability. During this period, the culture medium was changed every 2–3 days, and an apical wash was performed every week to eliminate the accumulated mucus. Any changes in the morphology of the cells, specifically protrusions on the MucilAir surface or perforations in its structure, were assessed under a light microscope (Olympus CKX41, Tokyo, Japan; 200× magnification), as was the presence of beating cilia. Typical microscopic images of the MucilAir^TM^ tissue and BEAS-2B cells grown at the ALI are shown in Figure 6.

### 4.2. Exposure to Complete Emissions—Technical Aspects

The exposure tests were conducted on a transient engine dynamometer on a Euro 5 direct injection spark ignition engine. The engine speed and either torque or accelerator pedal position (depending on the operating point) were programmed to match those experienced by the same model engine in a typical European middle class passenger car (Škoda Octavia) during a World Harmonized Light Vehicle Test Cycle (WLTC) driven on a chassis dynamometer. The tested fuel was a mixture of ordinary gasoline (BA-95N, Čepro, 4.9% ethanol, 0.3% ETBE) with ethanol to a final ethanol content of 20% ethanol (*v/v*). This fuel was denoted E20.

A proportional sampling gravimetric system was used to dilute the exhaust with filtered air at a constant dilution ratio of 10:1. and to sample the diluted exhaust onto 70-mm diameter fluorocarbon-coated glass filters (PallFlex, Pall, Portsmouth, UK) for particulate matter extraction (see below) and use of extracts in future studies. Prior to the sample reaching the filter, a portion of the sample was extracted into a toxicological incubator [41] in which an exposure box was located. The exhaust dilution was selected based on the results of pilot experiments [28]. To reach the desired conditions of 5% CO_2_, over 85% relative humidity and 37 °C in the incubator, CO_2_ was metered into the sample, and the sample was humidified by a selective membrane (Nafion, model no. FC125-240-5MR, PermaPure, Lakewood, NJ, USA) and routed into the in-house made exposure box containing a standardized 24-well plate, of which eight wells were populated with inserts with a 6-mm membrane to house the tested culture. The sample was distributed by symmetrical channels, at 25 cm^3^ per minute per channel, among the inserts, allowing “natural” (unassisted by thermophoresis, electrical charge, impaction, or other means) deposition by diffusion. The sampling system and the exposure chamber are described in detail in [28]. The constant 10:1 dilution ratio approach, while not being the most representative of the varying exhaust flow, was, so far, the most widely used approach in exhaust toxicity studies (a detailed discussion is given in [28]).

The mean concentrations of particulate matter in the diluted exhaust, as determined by gravimetric analysis, were 0.05 mg/m^3^. The mean concentration of black soot, as measured by a photo-acoustic analyzer (AVL Microsoot Sensor, AVL List GmbH, Graz, Austria), was 0.02 mg/m^3^. Some particles, about one-third by count, were lost in the diffusion membrane humidifier; low flow through the humidifier and high humidity of the outgoing sample made it impossible to avoid losses. It is estimated that most of the losses can be attributed to the diffusion of very small particles, and that the losses of the mass basis are relatively small. It is also estimated that the deposition rate of particles by diffusion is about 2% [28]. Therefore, the particle loading is estimated to be 3 mg per insert.

The filters with collected particulate matter were extracted with 60 mL of dichloromethane and 3 mL of cyclohexane for 3 h, extractable organic matter was pooled, and aliquots were used for chemical analysis. A detailed quantitative chemical analysis of PAHs and their derivatives was performed by HPLC with fluorimetric detection. The results are reported in Appendix A.

### 4.3. Exposure Scheme

The aim of this study was to mimic actual exposure as realistically as possible. A large portion of emissions usually originates from relatively short periods of exposure to less favorable conditions; typically associated with cold engine starts and highly dynamic operations. For these reasons, we opted for a dynamic driving cycle starting with a cold start. The WLTC, encompassing a range of operating conditions and currently used in the EU as a part of the type approval procedure was run twice - once with a cold start and once with a hot start. The engine was then actively cooled for a period of two hours, during which the cells were exposed to clean air and the sequence of two WLTC, one with cold start and one with hot start was repeated. This combination of conditions was denoted as “the exposure unit”. The control samples were exposed to filtered clean ambient air. The exposure was performed in a recently developed exposure system [28]. Two exposure schemes were used (Figure 7). For one-day exposure, TEER was assessed and medium collected for further analyses of mucin, adenylate kinase and lactate dehydrogenase (time point T0), before exposure to complete emissions. The cells were then transported to the exposure facility, where the exposure unit commenced. During the transportation, the cells were kept at 37 °C and sealed to avoid drying and loss of CO_2_-enriched atmosphere. Once the exposure unit had completed, the cells were transported back to the laboratory, TEER was measured, and the culture medium and cells were collected and stored at −80 °C for further analyses (time point T1) (Figure 7A). For the five-day exposure, the cells were first treated identically to the one-day treatment apart from time point T1 in which the cells were not collected. Instead, the medium was stored for laboratory analyses, replaced with a fresh one and the cell cultures were grown in the cell incubator at standard conditions overnight. The following day, the entire process was repeated, except for the steps at T0 that were omitted. After three more repeats, during which the culture medium was obtained at the end of the exposure units (time points T2–T4), the culture medium and cells were collected (time point T5) and the exposure was completed (Figure 7B). This combination of conditions allowed us to consider the periods of exposure to high concentrations of air pollutants (a stay on busy streets), background levels of air pollution (low traffic streets/indoors) as well as to include recovery periods (overnight incubations) for five-day exposures.

### 4.4. Microscopy Analysis of the MucilAir^TM^ Tissues and BEAS-2B Cells

The tissue inserts were fixed with a formalin solution (neutral buffered, 10%) at 4 °C overnight, then washed twice with PBS, and subsequently gradually washed with 25%, 50%, and 70% ethanol and embedded in paraffin. Paraffin-embedded tissues were cut into 6-µm sections, de-paraffinized, and the rehydrated sections were stained with hematoxylin–eosin. The images were recorded using an Olympus BX41 (Tokyo, Japan) light microscope in final magnification 1000×.

### 4.5. TEER Assessment and Mucin Analysis

For TEER measurements, an EVOM2 ohm meter (World Precision Instruments, Sarasota, FL, USA) in combination with an STX2 electrode was used [25]. This method represents a quantitative nondestructive way of measuring the integrity of tight junction dynamics in cell culture models. The TEER measurements were taken prior to (T0) and after every exposure to the complete emissions (T2–T5) (Figure 7). Resistance values were calculated according to the formula:$$\begin{array}{l} \mathrm{TEER}\;\left({Ω}^{*}{\mathrm{cm}}^{2}\right)\\ =\;\left(\mathrm{resistance}\;\mathrm{of}\;\mathrm{the}\;\mathrm{test}\;\mathrm{tissue}\;\left(Ω\right)\;-\;\mathrm{resistance}\;\mathrm{value}\;\mathrm{of}\;\mathrm{the}\;\mathrm{untreated}\;\mathrm{membrane}\;\left(Ω\right)\right)\;\\ \times \;\mathrm{surface}\;\mathrm{area}\;\mathrm{of}\;\mathrm{the}\;\mathrm{epithelium}\;\left({\mathrm{cm}}^{2}\right) \end{array}$$
where the resistance value of the untreated membrane is 100 Ω and the surface of the epithelium is 0.33 cm^2^.

Mucin production is used as a general marker of airway damage. For its analysis, a sandwich enzyme-linked lectin assay (ELLA) developed by Epithelix Sàrl (Geneva, Switzerland) was used. Briefly, a 96-well plate was coated with lectin from Triticum vulgaris and incubated for 1 h at 37 °C. After washing the plate, samples (apical medium, 50 µL/well) were added and incubated for 30 min at 37 °C. After another washing step, 50 µL/well of detection solution (glycine max soybean lectin–horseradish peroxidase conjugate) were pipetted into the wells and the plate was incubated for 30 min at 37 °C. The color was developed after washing and adding the TMB substrate (100 µL/well). The reaction was stopped by H_2_SO_4_ (50 µL/well) and the absorbance was measured at 490 nm, using a spectrophotometer (SpectraMax^®^ M5e; Molecular Devices, San Jose, CA, USA).

### 4.6. Cytotoxicity Determination

Cytotoxicity was measured prior to exposure (T0) and after every exposure unit (T2–T5) (Figure 7) in a basal culture medium using two commercial kits. Lactate dehydrogenase was assessed using the Cytotoxicity Detection Kit (Roche, Basel, Switzerland) and adenylate kinase, using the Adenylate Kinase Cytotoxicity Assay Kit (Abcam, Cambridge, UK). Both assays measure the activity of enzymes leaked from damaged cells. The analyses were performed according to the manufacturer’s instructions. The results were presented as the percentage cytotoxicity relative to the value obtained from the positive control (a 1-h exposure to 1% *v*/*v* Triton X-100 in culture medium at 37 °C).

### 4.7. Phosphorylation of Histone H2AX

The induction of double-strand DNA breaks was assessed by the detection of serine 139 phosphorylation of histone H2AX. The levels of phosphorylated H2AX (γ-H2AX) were measured after exposure to complete emissions (at time point T1 or T5, for the one-day and five-day exposure, respectively; Figure 7) in cell lysates containing 5 µg of total proteins using the commercial ELISA kit (HT γ-H2AX Pharmacodynamic Assay; Trevigen, Gaithersburg, MD, USA). The assay was performed according to the manufacturer’s instructions; H2AX phosphorylation was expressed in pM based on the standard curve generated using the γ-H2AX standard provided with the kit.

### 4.8. mRNA Expression Analysis

mRNA expression was assessed after exposure to complete emissions, at time point T1 or T5, for the one-day and five-day exposure, respectively (Figure 7). Total RNA from MucilAir^TM^ and BEAS-2B cells was isolated using NucleoSpin RNA XS kit (Macherey-Nagel, Düren, Germany) according to the manufacturer´s instructions. First, the inserts were washed with PBS containing Ca^2+^/Mg^2+^ (500 µL/well for basolateral washing and 200 µL on apical surface of the insert for apical washing). A mixture of 100 µl Buffer RA1 and 2 µL TCEP (NucleoSpin RNA XS) was added to the apical surface of each insert. Cells were scraped with 200 µL tip, homogenized by vigorous vortexing and RNA extraction proceeded according to the instruction manual. RNA concentration was quantified with a Nanodrop ND-1000 spectrophotometer (Thermo Fisher Scientific, Wilmington, DE, USA).

Human Molecular Toxicology Transcriptome panel (Qiaseq Targeted RNA Panel, Qiagen, Hilden, Germany) was selected for targeted mRNA expression analysis. The panel consists of 398 key genes (a list is available in Appendix A) playing a role in apoptosis, necrosis, DNA damage and repair, mitochondrial energy metabolism, fatty acid metabolism, oxidative stress and antioxidant response, heat shock response, endoplasmic reticulum stress and unfolded protein response, cytochrome P450s and phase I drug metabolism, steatosis, cholestasis, phospholipidosis, and immunotoxicity.

Libraries were prepared according to the manufacturer’s instructions. Briefly, 400 ng of RNA was treated by DNA elimination reagents and reverse transcribed into cDNA. One µL of cDNA was used for the molecular barcoding step, including the molecular barcoded gene-specific primer (GSP1) in a multiplex primer panel. Barcoded cDNA was purified over QIAseq Beads and a PCR reaction was prepared with a second pool of gene-specific primers (GSP2) and RS2 primer [95 °C 15 min; 8 cycles (95 °C, 15 sec; 60 °C, 5 min)]. cDNA was purified again, and a universal PCR was set up with RS-D and FS-D index primers for differentiation of each sample ((95 °C, 15 min; 20 cycles (95 °C, 15 s; 60 °C, 2 min)). A final clean-up was performed, and libraries were qualified and quantified. The library concentration was determined by Qubit 1× dsDNA HS Assay Kit on Qubit 4 fluorometer (Thermo Fisher Scientific, Wilmington, DE, USA). Libraries were validated on the Fragment analyzer system (Agilent Technologies, Santa Clara, CA, USA) using HS NGS Fragment Kit. The sequencing of libraries was performed on the NextSeq system (Illumina, San Diego, CA, USA).

NF-CORE RNASeq pipeline (https://github.com/nf-core/rnaseq, version 1.3) [42] was used to analyze RNA sequencing data. It pre-processed raw data from FastQ inputs (FastQC, Trim Galore!), aligned the reads (STAR), generated gene counts (featureCounts), and performed extensive quality-control on the results (RSeQC, dupRadar, Preseq, edgeR, MultiQC). The reads were mapped to a reference genome GRCh38.p12. DESeq2 [43] with default parameter settings was applied to normalize read counts and to identify differences in gene expression between sample groups. A multiple testing correction was performed using the Benjamini and Hochberg method. To account for gender differences of subjects from which the MucilAir^TM^ tissues and BEAS-2B cells originated, all genes located on chromosome X were omitted in gene expression analyses.

### 4.9. Statistical Analysis

The cell cultures were exposed in biological triplicates/parameter. For the laboratory analysis of mucin, AK, LDH, and γ-H2AX, technical duplicates of each biological replicate were tested. TEER values were obtained from a total of nine inserts/sample. The parameters were compared using two-way ANOVA with Sidak’s (post hoc) multiple comparison test and using Student’s *t*-test (GraphPad Prism version 8 (GraphPad Software Inc., San Diego, CA, USA)). Data were expressed as mean ± standard deviation (SD). Significance values ≤ 0.05 were considered significant. Venn diagrams were prepared in the Bioinformatics & Evolutionary Genomics tool (http://bioinformatics.psb.ugent.be/webtools/Venn/).

## Figures and Tables

**Figure 1 ijms-20-05710-f001:**
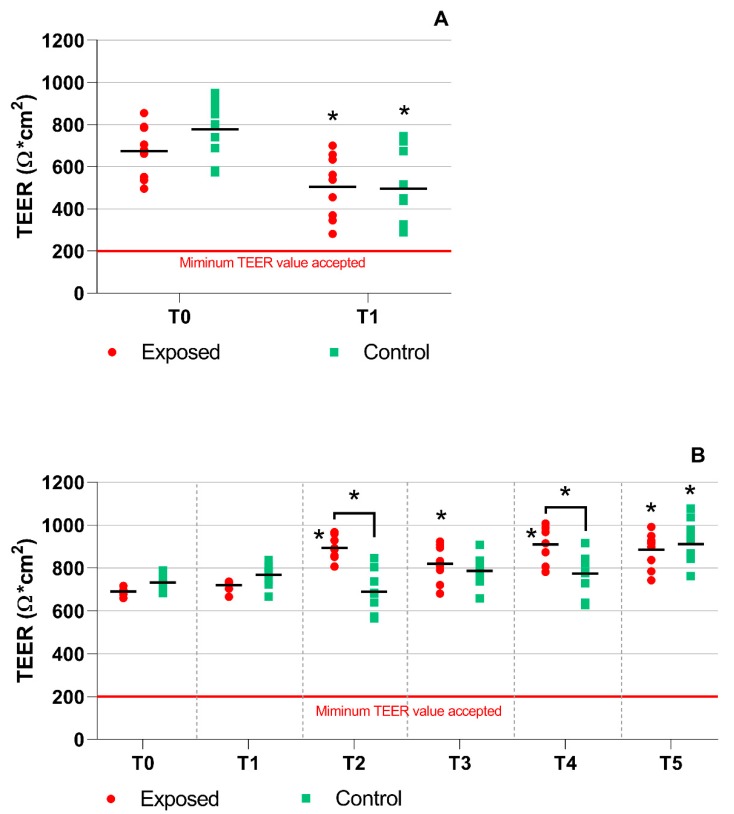
Transepithelial electrical resistance (TEER) measurement in the MucilAir^TM^ tissues. (**A**) A significant drop in TEER values was observed after one-day exposure. (**B**) A significant increase of TEER values in exposed samples after longer exposures (time points T2–T5). Asterisks denote significant (*p* ≤ 0.05) differences between T0 and later time points or between the exposed and controls for a given time point (T2 and T4).

**Figure 2 ijms-20-05710-f002:**
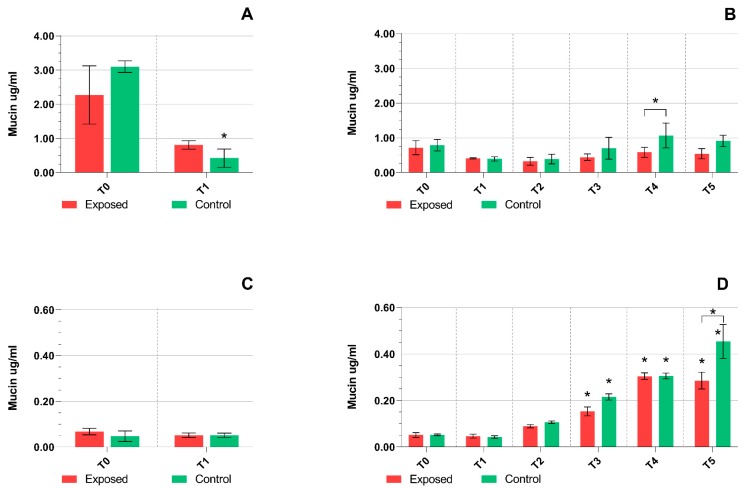
Mucin production by BEAS-2B cells and the MucilAir^TM^ tissues. (**A**) A decrease in mucin production by the MucilAir^TM^ tissues after one-day treatment. (**B**) A weak increase of mucin levels after longer treatment of the MucilAir^TM^ tissues (time points T3–T5). (**C**) Low mucin production in BEAS-2B cells after one-day exposure. (**D**) Time-dependent increase of mucin production in BEAS-2B cells after longer exposure periods. Asterisks denote significant (*p* ≤ 0.05) differences between T0 and later time points or between the exposed and controls for a given time point (T4 and T5).

**Figure 3 ijms-20-05710-f003:**
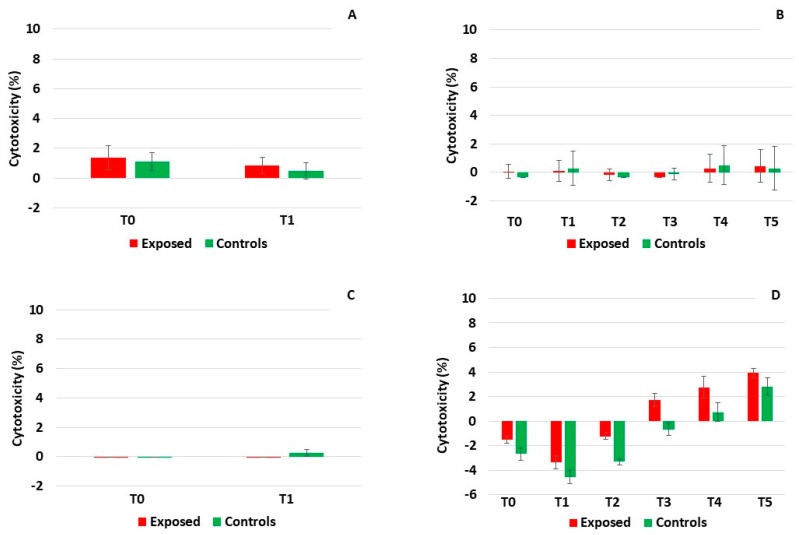
The activity of adenylate kinase after exposure to complete emissions. Very low activity in the MucilAir^TM^ system after one-day (**A**) and five-day (**B**) exposure. (**C**) Minimal activity in BEAS-2B cells after one-day exposure. (**D**) A weak response in BEAS-2B cells after five-day treatment (time points T3–T5).

**Figure 4 ijms-20-05710-f004:**
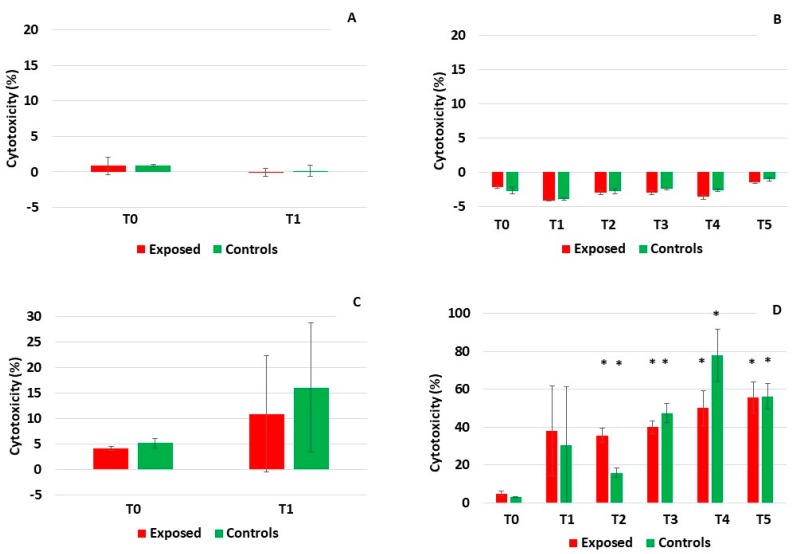
The activity of lactate dehydrogenase after exposure to complete emissions. No activity in the MucilAir^TM^ system after one-day (**A**) and five-day (**B**) exposure. (**C**) Increased LDH activity in BEAS-2B cells after one-day exposure. (**D**) High LDH activity in BEAS-2B cells after five-day treatment in both exposed and control samples. Asterisks denote significant (*p* ≤ 0.05) differences between T0 and later time points.

**Figure 5 ijms-20-05710-f005:**
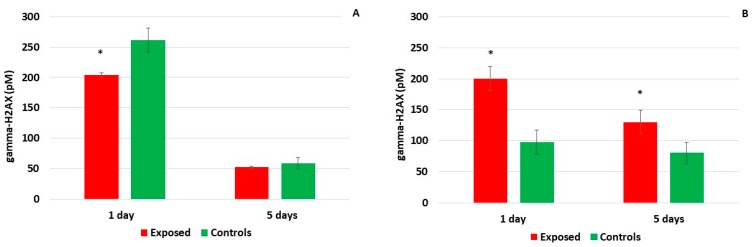
Histone H2AX phosphorylation after exposure to complete emissions. (**A**) No induction in the MucilAir^TM^ system after either time period. (**B**) Increased levels of DNA double-strand breaks in BEAS-2B cells after one-day and five-day treatment. Asterisks denote significant (*p* ≤ 0.05) differences between the exposed cells and controls.

**Figure 6 ijms-20-05710-f006:**
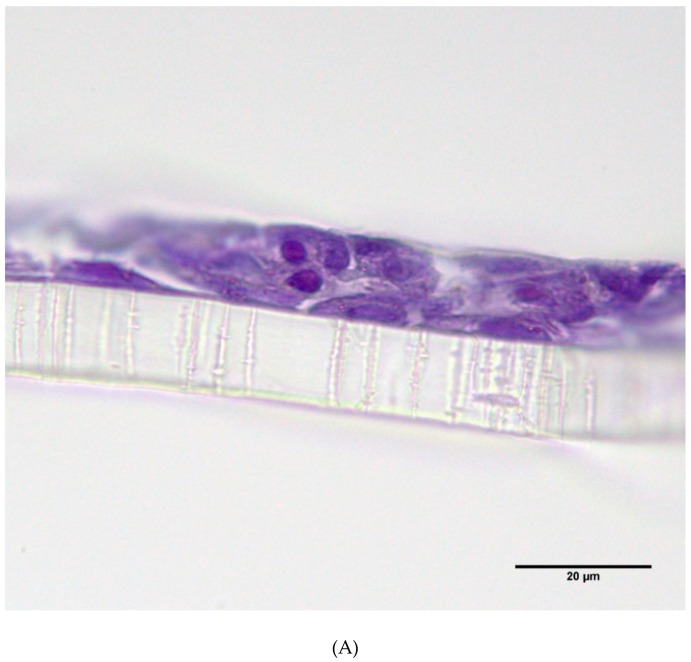
Microscopic images of cell models grown at the air–liquid interface (ALI) used in the study. The images were recorded using an Olympus BX41 light microscope using final magnification 1000×. (**A**) MucilAir^TM^ tissue grown on polyester membrane (0.4-µm pore size) with visible cilia on the apical side of the tissue, (**B**) BEAS-2B cells grown at ALI conditions (polyester membrane (0.4-µm pore size)) formed a monolayer with no visible over-grown structures.

**Figure 7 ijms-20-05710-f007:**
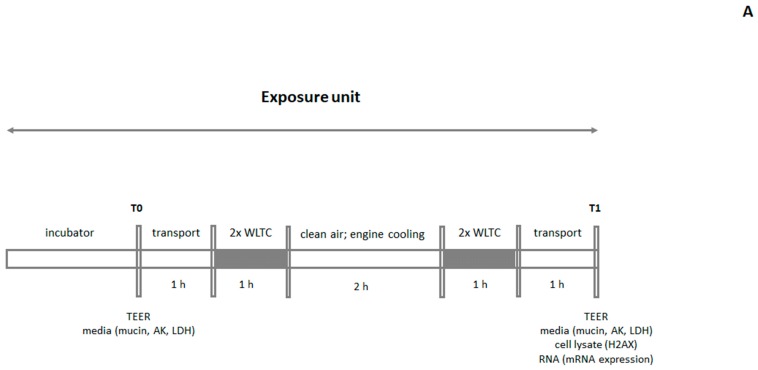
Exposure schemes used to expose the MucilAir^TM^ tissues and BEAS-2B cells to complete emissions. (**A**) One-day exposure and definition of “the exposure unit”. (**B**) Five-day exposure. Individual time points (T0–T5) and parameters analyzed at these time points are shown. For details, see the description in Section 4.3. WLTC: World Harmonized Light Vehicle Test Cycle; AK: adenylate kinase; LDH: lactate dehydrogenase.

**Table 1 ijms-20-05710-t001:** The expression of genes induced by exposure to complete emissions in the MucilAir^TM^ tissues.

Exposure	Gene Name	Ensembl ID	Biological Pathway	Log2 FC	Adj. *p*-Value
**One day**	*HSPA5*	ENSG00000044574	Heat shock response	–0.877	0.001
	*CYP1A1*	ENSG00000140465	Cytochrome P450s and phase I drug metabolism Immunotoxicity	2.634	0.042
**Five days**	*CYP1A1*	ENSG00000140465	Cytochrome P450s and phase I drug metabolism Immunotoxicity	3.060	0.007

**Table 2 ijms-20-05710-t002:** The expression of genes induced at time point T5 when compared with T1 in the exposed MucilAir^TM^ tissues.

Gene Name	Ensembl ID	Biological Pathway	Log2 FC	Adj. *p*-Value
*SLC2A3*	ENSG00000059804	Phospholipidosis	–3.008	0.025
*AQP4*	ENSG00000171885	Steatosis	–1.140	0.027

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
