# Peer review of "The Biological Effects of Complete Gasoline Engine Emissions Exposure in a 3D Human Airway Model (MucilAirTM) and in Human Bronchial Epithelial Cells (BEAS-2B)"

_ijms, 2019, doi:10.3390/ijms20225710_

Round 1

Reviewer 1 Report

The manuscript by Rossner and colleagues describes toxicological investigations in an air liquid exposure system in which either human primary bronchial epithelium (MucilAir) or a human bronchial epithelial cell line BEAS2B were exposed to exhaust from a gasoline engine. Besides typical assays for cytotoxicity, gene expression analyses of exposed and non-exposed cells were performed. In general the effects of 1 day or up 5 days exposure (2 1h-intervals per day) did not reveal dramatic toxicological consequences.

The development of in vitro exposure systems in which effects of volatile and particulate xenobiotics on the airways can be investigated is a highly relevant topic in modern toxicology. There are some well characterized air liquid exposure systems available already. In the current manuscript, the authors seem to use a kind of home made system which is not well described in the text. Due to the importance of the approach and because of the soundness of the performed assays I think that the data should be published. However, I have some concerns which mainly address the question how relevant this in vitro system is:

First, the exposure should be somehow characterized with respect to physical and chemical properties. Are there chemical analyses available for the exhaust or for sampled particles? What is the real concentration of pollutants on the cells? Why is the exhaust diluted by 1:10? Other systems at least offer the opportunity to determine the mass of deposited particles. Is this also possible for the described system?

The authors describe only weak effects of gasoline engine exhaust in the used cells. They suggest that the exhaust of modern gasoline engines is of rather low toxicity for humans. However, would they be able to detect toxic effects at all in their system? This is of particular importance when a new experimental system is established and used. They did not include positive controls in their experiments. Maybe there are data with diesel exhaust available which can be correlated to known human health effects. This should then be mentioned here.

In most of the graphs it is not clear for me which significances are indicated by the asterisks. Do they indicate differences between exposed and control or between T0 and later time points?

Author Response

The manuscript by Rossner and colleagues describes toxicological investigations in an air liquid exposure system in which either human primary bronchial epithelium (MucilAir) or a human bronchial epithelial cell line BEAS2B were exposed to exhaust from a gasoline engine. Besides typical assays for cytotoxicity, gene expression analyses of exposed and non-exposed cells were performed. In general the effects of 1 day or up 5 days exposure (2 1h-intervals per day) did not reveal dramatic toxicological consequences.

The development of in vitro exposure systems in which effects of volatile and particulate xenobiotics on the airways can be investigated is a highly relevant topic in modern toxicology. There are some well characterized air liquid exposure systems available already. In the current manuscript, the authors seem to use a kind of home made system which is not well described in the text. Due to the importance of the approach and because of the soundness of the performed assays I think that the data should be published. However, I have some concerns which mainly address the question how relevant this in vitro system is:

First, the exposure should be somehow characterized with respect to physical and chemical properties. Are there chemical analyses available for the exhaust or for sampled particles?

Response: Chemical analysis of organic compounds (polycyclic aromatic hydrocarbons, PAHs, and their derivatives) in particulate matter collected on the filters has been performed. The data is presented in Supplementary File 5. Information on chemical analysis was added to Materials and Methods.

What is the real concentration of pollutants on the cells?

Response: This analysis was not performed in our study. We assessed the concentration of particulate matter, black soot and organic compounds (PAHs) in the diluted exhaust, but information of the pollutants reaching the cells is not available. The data on concentration of particulate matter and black carbon was added to Materials and Methods.

Why is the exhaust diluted by 1:10?

Response: This is a dose selected based on our pilot experiments in which we looked for exposure conditions that would induce biological response but would not be cytotoxic. The problem was that sensitivity and response of the cellular models differed: BEAS-2B cells were more sensitive than the MucilAir tissues and biological effects in these cells were more pronounced. For MucilAir tissues longer exposure periods and/or lower dilution (i.e. 1:5) would have been better, but these conditions were not optimal for BEAS-2B cells due to cytotoxicity. Therefore, we had to select treatment acceptable for both models. Exhaust dilution 1:10 proved to be a compromise. The explanation is given in the text (Chapter 4.2).

Other systems at least offer the opportunity to determine the mass of deposited particles. Is this also possible for the described system?

Response: We have information on particle concentration in the diluted exhaust and based on information presented in our previous study [1] we estimated that about one third of particles by count were lost in the diffusion membrane humidifier. The deposition rate of particles by diffusion was about 2% [1]. We thus estimate the particle loading to be 3 mg per insert. This information is now provided in Materials and Methods.

The authors describe only weak effects of gasoline engine exhaust in the used cells. They suggest that the exhaust of modern gasoline engines is of rather low toxicity for humans. However, would they be able to detect toxic effects at all in their system? This is of particular importance when a new experimental system is established and used. They did not include positive controls in their experiments. Maybe there are data with diesel exhaust available which can be correlated to known human health effects. This should then be mentioned here.

Response: Yes, we were able to induce toxic effects when lower dilution of exhaust was used. In the pilot experiments we tested several dilutions (1:5, 1:10, 1:20) and exposure schemes. We aimed to find the conditions that induced low cytotoxicity, but at the same time had some biological effect. As the response of BEAS-2B cells and the MucilAir system differed (BEAS-2B cells were more sensitive than the 3D model), we had to find a compromise suitable for both model systems. While we were not able to find any study correlating the effects of modern engine exhaust (either gasoline, or diesel) on human health, in a study in rats no adverse effects of 7-days and 28-days inhalation of modern diesel engine exhaust was found (Magnusson et al., Inhal Toxicol. 29 (2017) 206–218) suggesting low toxicity also for humans.

In most of the graphs it is not clear for me which significances are indicated by the asterisks. Do they indicate differences between exposed and control or between T0 and later time points?

Response: The significance is primarily indicated for a comparison between T0 and later time points. The difference between exposed and control samples within a certain time point is graphically indicated in the figure. In Figure 5, the comparison was between exposed and controls. Figure legends were corrected so that the indication of significance in the graphs was clear.

Reviewer 2 Report

The study was well designed, the paper is well written with detailed explanations. The exposure protocol is interesting even if it is difficult to understand for non specialists (see below)

Main points

1/ In the introduction to this study, the authors highlight the importance of conducting studies on the long-term effects of exposure to complete emissions. However, the duration of exposure to the mucilair model is 5 days, it would have been more interesting to choose a longer exposure time (28 days).

2/ The authors demonstrate that BEAS-2B is not suitable for exposure at the ALI and that the toxic effects are due to the exposure protocol rather than by exposure to the toxic. This should be clearly indicated in the abstract.

3/ In addition, toxic effects on BEAS-2B are mainly induced by culture and exposure conditions rather than by exposure to the toxic (LDH, mRNA analysis). As a result, tables 3A and 3B have little interest and should be deleted or added to supplementary material. In addition, the results of Figure 6 are given in the text and this figure can be deleted.

4/ Figure 7 is difficult to understand, and does not provide information on the toxic effect studied. I suggest that it be deleted as well.

5/ The exposition schemes can be simplified: 1 schema for T1 exposure and another for T1-T5 exposure

You can simply indicate T1,2,3,4 and 5 below next to the name of the tests performed

Moreover WLTC is not in the abbreviations, please write in full, to facilitate understanding.

6/ The protocol (chapter 4.2) is not clear for me, see below. Could you explain?

“ The exhaust was diluted at a constant dilution ratio of 10:1 and routed into a toxicological incubator [41].”

“The sampling system and the exposure chamber are described in detail in [28].”

How did you check that the exposure was effective?, please explain

Minor points

L359, mistake with the second “the”?

Author Response

The study was well designed, the paper is well written with detailed explanations. The exposure protocol is interesting even if it is difficult to understand for non specialists (see below)

Main points

1/ In the introduction to this study, the authors highlight the importance of conducting studies on the long-term effects of exposure to complete emissions. However, the duration of exposure to the mucilair model is 5 days, it would have been more interesting to choose a longer exposure time (28 days).

Response: We agree with the Reviewer that longer exposure times would be more interesting. We were considering longer exposure periods, but such scheme would be logistically very demanding (operating and maintaining the exposure system is time consuming and requires skilled workers that would not be available for extended time period). Moreover, we aimed to compare the response of the standard cell culture (BEAS-2B cells) with the 3D model. However, the BEAS-2B cells cannot be grown longer than couple of days and then they would require passaging which would change the properties of the cell culture. We thus selected the period acceptable given the circumstances. In any case, longer exposure of the MucilAir system in future studies would be very interesting. We added the comment to the last paragraph of Discussion.

2/ The authors demonstrate that BEAS-2B is not suitable for exposure at the ALI and that the toxic effects are due to the exposure protocol rather than by exposure to the toxic. This should be clearly indicated in the abstract.

Response: The abstract was corrected to indicate the effect of the exposure protocol.

3/ In addition, toxic effects on BEAS-2B are mainly induced by culture and exposure conditions rather than by exposure to the toxic (LDH, mRNA analysis). As a result, tables 3A and 3B have little interest and should be deleted or added to supplementary material. In addition, the results of Figure 6 are given in the text and this figure can be deleted.

Response: We agree that the data in Table 3 and Figure 6 are related to cultivation conditions rather than to the effect of emissions. As suggested, Table 3 was added to the supplementary material (Supplementary File 1) and Figure 6 was deleted.

4/ Figure 7 is difficult to understand, and does not provide information on the toxic effect studied. I suggest that it be deleted as well.

Response: We agree that the data in this figure is not directly related to the effect of engine exhaust. We still suggest keeping it in the supplemental material (Supplemental File 3), as it illustrates relationships between genes deregulated as a result of exposure and in the controls.

5/ The exposition schemes can be simplified: 1 schema for T1 exposure and another for T1-T5 exposure. You can simply indicate T1,2,3,4 and 5 below next to the name of the tests performed

Response: The scheme was simplified as suggested.

Moreover WLTC is not in the abbreviations, please write in full, to facilitate understanding.

Response: The abbreviation was added to the list.

6/ The protocol (chapter 4.2) is not clear for me, see below. Could you explain?

“ The exhaust was diluted at a constant dilution ratio of 10:1 and routed into a toxicological incubator [41].”

“The sampling system and the exposure chamber are described in detail in [28].”

Response: The exposure chamber, a box in which a 24-well plate with Transwell inserts was put, was developed by our team and is described in detail by Vojtisek-Lom et al. (SAE Technical Paper 2019-24-0050. (2019). https://www.sae.org/publications/technical-papers/content/2019-24-0050/. (ref. [28]). The chamber was located in a toxicological incubator described in ref. 41, that was originally developed for exposure of lung tissue slices and was modified by our team to be used for our experiments. The text in the manuscript was modified.

How did you check that the exposure was effective?, please explain

Response: As mentioned in the response to Reviewer 1, we measured the concentration of particulate matter, black soot and organic compounds (PAHs) in the diluted exhaust and estimated particle deposition to be 3 mg/insert. The exact measurement of chemicals and/or particles reaching the cells was not performed. However, the effectiveness of the exposure can be judged by assessment of biological parameters (cytotoxicity detection, mucin production, DNA damage, gene expression changes), that were changed in response to treatment, although the changes were often weak.

Minor points

L359, mistake with the second “the”?

Response: Corrected.

Reviewer 3 Report

The authors report the in vitro effects of complete gasoline engine emissions exposure using primary cell based human epithelium and bronchial cell line.

Although modest biological effects have been reported, the described experimental approach is novel. The article is well written and deserve publication.

Author Response

The authors report the in vitro effects of complete gasoline engine emissions exposure using primary cell based human epithelium and bronchial cell line.

Although modest biological effects have been reported, the described experimental approach is novel. The article is well written and deserve publication.

Response: Thank you for the comment. The manuscript was modified as requested by other Reviewers.